# Key Concepts, Weakness and Benchmark on Hash Table Data Structures

**Santiago Tapia-Fernández** * , **Daniel García-García and Pablo García-Hernandez**

Escuela Técnica Superior de Ingeniería Informática, Universidad Politécnica de Madrid (UPM), Calle Ramiro de Maeztu, 7, 28040 Madrid, Spain; daniel.garcia.garcia@alumnos.upm.es (D.G.-G.); pablo.garcia.hernandez@alumnos.upm.es (P.G.-H.)
* Correspondence: santiago.tapia@upm.es

**Abstract:** Most computer programs or applications need fast data structures. The performance of a data structure is necessarily influenced by the complexity of its common operations; thus, any data-structure that exhibits a theoretical complexity of amortized constant time in several of its main operations should draw a lot of attention. Such is the case of a family of data structures that is called hash tables. However, what is the real efficiency of these hash tables? That is an interesting question with no simple answer and there are some issues to be considered. Of course, there is not a unique hash table; in fact, there are several sub-groups of hash tables, and, even more, not all programming languages use the same variety of hash tables in their default hash table implementation, neither they have the same interface. Nevertheless, all hash tables do have a common issue: they have to solve hash collisions; that is a potential weakness and it also induces a classification of hash tables according to the strategy to solve collisions. In this paper, some key concepts about hash tables are exposed and some definitions about those key concepts are reviewed and clarified, especially in order to study the characteristics of the main strategies to implement hash tables and how they deal with hash collisions. Then, some benchmark cases are designed and presented to assess the performance of hash tables. The cases have been designed to be randomized, to be self-tested, to be representative of a real user cases, and to expose and analyze the impact of different factors over the performance across different hash tables and programming languages. Then, all cases have been programmed using C++, Java and Python and analyzed in terms of interfaces and efficiency (time and memory). The benchmark yields important results about the performance of these structures and its (lack of) relationship with complexity analysis.

**Keywords:** data structures; hash table; hash tree; algorithm performance; complexity analysis

## 1. Introduction

Associate arrays or maps are a common data structure in programs. They provide support to establish relations between data and, since they are expected to be faster than other alternatives, they have became a common choice among programmers in any domain and programming languages. Therefore, studying the real performance of these data structures and knowing some of their problems and limits is an interesting matter for a lot of users and potential users.

The objective of this paper is to study the conceptual base of hash tables and then assess and analyze the standard implementations of hash table data structures. Since the theory about hash tables is well-known and established, the emphasis is not going to be the data structures and algorithm design (of course, we summarize it to provide context). The goal is to study the actual (standard) implementations of hash tables from an empirical point of view, that is, the emphasis will be running real programs that use hash tables in real computers and measure program executions in terms of time and memory use. As a consequence, manuals or documentation will be used as bibliography sources since the

focus is on the implementation and their details and not on theoretical designs. We will use the theoretical context to explain why the assessment programs do what they do and to *try* to explain the performance results, but an exhaustive review of bibliography is not intended. On summary, our methodology approach to the problem follows a **black box** analysis strategy.

The hash tables will be used without seeking optimizations. That means that the improvements or optimizations in using the hash tables will be avoid even if we know about them. For instance, most implementations could improve their building (insertion) step by providing the expected size, but it is avoided, since the objective is using the data structures as if no relevant information about the program requirements is known in advance. Even more, since the goal is to measure time and memory use, program profiling is also avoided because it could introduce some overload (even if minimal) and because the objective is not improving the programs but just to measure their behavior.

Finally, since there are just a few standard implementation of hash tables in the chosen programming language, some other alternatives of hash table algorithms have been selected and implemented to provide more data for analysis. The selectiondoes not have a specific criteria nor does it look for any particular feature, just that the information to implement them is widely available and is clear enough to write an implementation easily and shortly. That is, the best algorithms were not actively sought.

### 1.1. Theoretical and Math Background

Many authors have described the *map* (also known as *associative array*) data structure, although those definitions differ in their detail and approach, as a general one we can defineit as an abstract data type that represents a correspondence between two sets, that is, given two arbitrary sets $\mathbb{K}$ and $\mathbb{V}$, $k \in \mathbb{K}$, $v \in \mathbb{V}$, a map represents a general correspondence $k \rightarrow v$. A map can also be interpreted as a set of pairs $\langle k, v \rangle$ that meet the condition that there are no pairs with the same $k$ values, i.e., $\forall p, p' \in \langle k, v \rangle, p \neq p' \implies k \neq k'$. For now on, $\mathbb{K}$ will be the set of *keys* and $\mathbb{V}$ the set of *values*.

Maps are translated into programming languages with interfaces that, basically, allow to: insert pairs, get or update values using the key as a parameter, and remove pairs also using the key as parameter. Different programming languages name this abstract data type and its implementations differently. Thus, C++ uses *associate container* as the name for the map *concept* (requirement or interface) [1] and `map` and `unordered_map` as the names of the main implementation (models) of that concept; Java has an interface `Map<K,V>` [2] for *maps* and several standard classes implementing that interface, for instance: `HashMap`, `TreeMap` or `Hashtable;` and Python has a standard class `dict` that implements the interface of a *map* [3], but note that there is no such a concept of *interface* in Python. Whatever the names, *map* implementations can be done using several families of data structures. These three are good examples: standard sequences, tree-like structures, and hash tables:

- **Standard sequences (or Direct-address tables)** Ref. [4]. Although they are seldom used, arrays or lists can be used as maps as long as they store pairs $\langle k, v \rangle$ as elements. Since the complexity of searching for a pair is $\mathcal{O}(n)$ and searching pairs is usually the main application when using maps, this is very uncommon.
- **Tree-like structures.** Balanced Binary Trees [4], AVL Trees [5], or Tries [6] are some alternatives of tree-like structures that could be used for implementing a map. Tree-like data structures have an additional requirement with respect to the other alternatives: they need a sorting operation on the key set, $\mathbb{K}$. Usually they have $\mathcal{O}(\log n)$ complexity for accessing elements; thus, although they are used in some applications they are being replaced by hash tables.
- **Hash tables** Ref. [4]. A hash table is basically an array containing the values or other data structures which can store chained values, such as linked lists, as in the `std::unordered_map` of the GCC implementation of the C++ standard library [7]. The indexes of the array are computed using a hash function. hash table structures have a remarkable property: most of their operations are (amortized, average, etc.) constant time, that is, their theoretical

complexity is $\mathcal{O}(1)$. Of course, such behavior is the most wanted characteristic in any context, and thus they became the object of our study.

*1.2. Concepts and Classification of Hash Tables*

Following the previous definitions, an array is a specialized map whose keys are consecutive nonnegative integers and values are any arbitrary type. Hash tables use an array as their base because accessing its elements by an index is classically a constant time operation. However, since we need a full map, there should be a method to produce integer indexes for arbitrary key sets and that is the role of the hash; in fact, the hash functions. Therefore, to define the hash table we need:

1. An application $h : \mathbb{K} \to \mathbb{Z}_{\geq 0}$, this is the *hash* function, it needs to be an application because any arbitrary $k \in \mathbb{K}$ needs a corresponding index to be stored in the array.
2. An array that maps: $\mathbb{Z}_{\geq 0} \to \mathbb{V}$. That array or the elements of the array in this particular context is usually named "*bucket* ".

In doing so, some issues arise:

- It is not possible (neither practical) to have an unlimited size array.
- Hash functions should distribute with "good uniformity" the images of the keys among the available integers, that is, within the range of indexes of the array.

These two issues produce:

- A new requirement: a new correspondence is needed in order to reduce the range of integers returned by the hash function. That is the real application between keys and indexes is the composition of $h : \mathbb{K} \to \mathbb{Z}_{\geq 0}$ and $h_r(N_b) : \mathbb{Z}_{\geq 0} \to \{0 \ldots N_b - 1\}$ where $\{0 \ldots N_b - 1\}$ is the target range of indexes, being $N_b$ the size of the array of *buckets*. The requirement of $h_r$ is seldom mentioned in the user manuals since it has no impact in the user interface. A simple implementation of this reduction is masking the original hash result to $2^{N_b} - 1$ which is equivalent to compute $h(k) \mod N_b$. The $h_r$ is called **range-hashing** (According to Tavori and Dreizin nomenclature, $h_r$, the ranged hashing has two parameters, but since the size of the bucket array is part of the state of the hash table object, it has been preferred to describe this function as having another side dependency, just as is $N_b$ were a constant), the composition of the hash function, $h$ and $h_r$ produces the **composed hash function** (Anew, in the Tavori and Dreizin document, there is not an specific name for this function composition. In their nomenclature, the ranged-hash function is defined as a whole not as a composition of two functions. Note that this ranged-hash function is not available for use in the standard implementation of hash tables (C++, Python, Java, ... ). The one available for custom implementation is the hash function), $h_c$, that solves the problem. That is, the composed hash function: $h_c = h_r(N_b) \circ h$, $h_c : \mathbb{K} \to \{0 \ldots N_b - 1\}$ transforms the elements of the key set into valid indexes of a given array. [8]
- A new concept: the collision, that is, the coincidence of the values of the composed hashing function of two different keys. There is a collision when $h_c(k_1) = h_c(k_2)$ where $k_1, k_2 \in \mathbb{K}$ and $k_1 \neq k_2$ [9]. Note that hash collisions are not the same as hash table collisions since there is an extra reduction step (in fact, $h_r$) that will necessarily produce more coincidences. Hereafter, these collisions will be denoted as **qualified collisions**.

In all hash table designs, the hash function is external to the hash table data structure, for instance, in Java all objects have an inherited method that returns a default hash for the object. In this context, hash functions should be carefully design to produce uniform distributed integers values from arbitrary data but other requirements for a cryptography hash are just not needed. Not surprisingly, in Java and C++ the standard implementation for integer keys is the identity since doing nothing is the fastest alternative and there are no collisions. Note that assuming this design implies that the hash function, $h$, is *external* while the reduction function, $h_r$, is *internal* to the data structure and should be considered as part of the algorithm. This is the design in all the programming languages in our study (and it is also present in others).

The alternative solutions for solving qualified collisions induce the classification of hash table data structures. Most implementations follow one of these two alternative strategies [9]:

- **Open Addressing.** When a *qualified collision* occurs another place is searched following a probing strategy, for instance: linear probing, double hashing, etc. The important feature is that at most one pair $\langle k, v \rangle$ is stored in the array (in the *bucket*).
- **Chaining**. When a qualified collision occurs, the pair is stored anyway in that array slot. That means that *buckets* do not store pairs $\langle k, v \rangle$ directly, but some other data structure capable of storing several pairs, usually linked lists, but also other data structures such as binary trees.

Even if most implementations could be classified either as open or chaining, some of them have their own particularities, for instance, cuckoo hashing follows an open address strategy, but it uses two hash tables instead of one as in the other alternatives [10].

### 1.3. Methodology

The work for this paper was planned as a survey or perspective, not as an actual research project, that is, no novelty was primarily intended. However, since the evaluation of empirical performance is an interesting matter, it was decided to focus the survey on the development of benchmark cases and the empirical evaluation of algorithms instead of focus on a theoretical comparison or review.

The first target of the study was finding the influence of the range-hashing function. As the default hash in Java and C++ for integer numbers is the identity function (a perfect hash function in terms of "good uniformity"), using integers as keys leaves an open window to look for the influence of the range-hashing function and, eventually, the influence of *qualified* collisions in the hash tables.

In fact, that influence was found, but the preliminary results also show something quite unexpected: the measured times were not constant with the data structure size. That is, the empirical measurements does not show the theoretical complexity $\mathcal{O}(1)$. That finding introduces a new target in the survey and its methodology because the goal from that moment was to confirm that fact by providing enough evidence.

Such evidence is based on:

1. A reliable measure of elapsed times. As any single operation on the standard mapping structures takes a very short time, it was decided to measure the time for millions of such operations. That way, measured times have a magnitude order of seconds. Since the absolute error in time measure should not exceed a few milliseconds, the relative error in the total time is very low.

2. Including checking operations in all benchmark cases. That is, benchmarks were designed as if they were unit test cases. That way, it is sure that they do what they are supposed to do. In addition, those checks avoid compiler optimizations (some compilers could avoid the evaluation of an expression in the source code if they detect that there is no further dependency on the resulting value). Benchmark cases are described in the next section.

3. Providing a wide variety of benchmark cases. That variety includes: using different programming languages, using standard and no standard implementations and using different value types as keys. In addition, of course, benchmark cases themselves are different. All of them share a similar design: they all build the data structure by inserting (a lot of) elements and they all execute a loop looking for keys. However, they do these general steps in different ways.

4. Evaluating other data structures that should be upper or lower bounds of performance. For instance, a binary tree could provide a lower bound since it has $\mathcal{O}(\log N)$ complexity, while an array could be an upper bound since it does not need doing any hash.

## 2. Materials and Methods

To evaluate performance, several benchmark cases have been written in Java, C++ and Python (Some other languages have been considered and used, but the benchmark cases and result analysis in those languages are not finished). These programming languages have been chosen because they have high rank at the TIOBE index of popularity and have support for generic programming. In fact, C language has been excluded in this study because it does not support generic programming.

All sources and materials needed to compile, run, and produce the data (and graphics) in this article are freely available for download or inspection at https://bitbucket.org/b-hashtree/b-hashtree/. Requirements to build or run the sources are minimum: a GNU/Linux distribution, a standard C++ compiler, a Java compiler and a Python 3 interpreter. For automatic building and execution *CMake* [11] should be used. To produce the data and graphics, it is also needed a bash shell and a LaTeX compiler with package *pgfplots* installed. All cases consist of:

- A program that runs operations on hash tables and other data structures (for comparison) written in at least Java, C++ and Python programming languages, using their standard or most common data structures to model a map. All benchmark cases have some trivial check in order to ensure that the operations are not optimized out and that the program actually does what it is supposed to do. All programs accept a parameter to pass the target size of the data structures (in $\log_2(size)$ for simplicity), that is, the number of elements to be inserted; most of them accept another parameter to select the actual data structure to run the case (when more than one option is available), and some also accept other numerical data to assess the influence of specific factors in the performance. Whenever possible, the number of evaluated operations is also a parameter and does not depend on the size. All programs output the elapsed time or speed (in loop-iterations per millisecond) for some steps in the program. These steps include, at least, a preparation step to generate and insert the elements into the map and another loop executing the operations to be evaluated. Each iteration includes the operation of interest and some other sentences, including conditional sentences for introducing some *branches* in the loops. Times are taken before and after running the loops using the standard function to access the system time in each language.

- Some *Bash* scripts and *CMake* configuration files (*CMakelist.txt*). Using *CMake* and the scripts it is possible to run automatically all the cases without manual manipulation. The result of the different targets is a collection of files (three files per benchmark case): the first one is the standard output of the program and includes the elapsed times and speed computed internally; the second is the output of the *GNU/Linux* command `/bin/time` (it is not the time command embedded in Bash) and the third is a logging file with the content from the error standard stream. The first two files are formatted as TSV files and their data will be used without manual manipulation into the TeX documents that generate the graphics. Since all source files, *CMakelist.txt* files and scripts could be read it is possible to review and validate the whole process. Of course, it is fully reproducible in any computer if the tools are installed.

- Some TeX documents that generate the graphics to be displayed. The graphics in this article are a selection among all the graphics that are generated, these graphics already generated were uploaded to the download area in the repository to facilitate casual access. In the graphical representation of results, speed is preferred to time, that way, the bigger are the values, the better is the performance. Note that the speed is computed in the number of iterations per millisecond and the iterations make some other operations apart from the map ones.

The benchmark cases have been developed for finding weaknesses and general performance. They are labeled in the sources using `CaseA`, `CaseB`, ... `CaseG`. Cases A and B were designed to find the influence of *qualified* collisions. Since C++ and Java use an identity function forthe integer hash function, these cases try to produce qualified collisions by computing integer keys with specific patterns. Case C seeks to determine if access-

ing and inserting/removing methods have similar performance. Since strings are often used as keys, Case D was designed to evaluate a very straightforward case of string use. Using not-so-obvious keys is reviewed in Case E, in this case, objects that represent three-dimension-float-number vectors are used as keys. Cases F, G are other cases of using string as keys, case F avoids using an array of keys to access the map and random generation of data; and case G represents cases where the pair $\langle k, v \rangle$ is bigger in terms of memory use.

Most cases are executed several times to modify the values of the parameters, including: problem size, hash table specific implementation, and other parameters. Typically, each case produces a little above of a hundred of data rows for C++ and Java and some dozens of row data for Python (there are less implementations in this language). A summary of the benchmark cases is shown at Table 1.

**Table 1.** Summary of Benchmark Cases.

| Case Id | $\mathbb{K}$ | $\mathbb{V}$ | Description | Motivation |
|---|---|---|---|---|
| A | 32-bit integer | double precision float | Keys are consecutive integers multiplied by a constant parameter ($\lambda$) and then shuffled before insertion. Values are random floats in [0,1]. Iteration computes the average of values, the expected result should be approximately 0.5. | Searching for ranging hash weaknesses and *qualified collision* resolution. |
| B | 32-bit integer | integer | Keys are consecutive integers multiplied by a constant parameter ($\lambda$) and then shuffled before insertion. Values are consecutive integers from 1 to Size. Iterations sum up the values and the result is checked with $\sum_{k=1}^{n} k = \frac{n(n+1)}{2}$. | Searching for ranging hash weaknesses and *qualify collision* resolution. |
| C | 32-bit integer | plain object | Keys are random integers. The values are objects storing different account attributes. Iterations either access, remove, or insert values into the map from and to a limited size queue. Number of operations are summed up for checking. | Comparing the performance of access vs. removal/insertion operations. All operations are evaluated around the assigned size. |
| D | string | double | Keys are strings generated from random integer numbers. Values and checking are as in Case A. | Assessing general performance while using string keys. |
| E | plain object | plain object | Keys are 3-dimension float vectors in a plain object. Values are plain objects storing: (a) an angle (float), (b) the index (integer) and (c) the count of access to the object (integer). Keys are generated along a helix curve using the parametric equations. Since the parameters are stored as values, each iteration computes anew points on the helix, checks equality, and counts the number of accesses. | Assessing performance on not-so-usual hash functions. Note that Java provides a default implementation. In Python, a tuple could be used for computing a default hash, but in C++ a custom hash is required. |
| F | string | plain object | Keys are strings. The values are designed to produce a new key, that is a new string. This way keys and values could be traversed, since elements are linked. Each iteration traverses the structure and count the number of accesses. Then the programs traverse the map once more and check the counting. | Assessing performance using strings once again, albeit this time there are no random numbers deliberately. |
| G | string | array of double floats | Keys are strings, the same as in case D, but twice longer. Values are arrays of 8 random floats from 0 to 1. Iterations compute the average of all numbers. Expected result is 0.5. | Assessing performance on bigger $\langle k, v \rangle$ pairs. As a collateral effect, there are some extra operations per iteration, since up to 8 floats have to be added per element. |

Apart from benchmark cases, some other well-known data structures were implemented from scratch. The goal of these developments is to complete a (numerous) set of alternative implementations, to have some reference for accepted high or low performance and to have equivalent data structure implementations across languages. They were developed from scratch to avoid external dependencies and to be fair about comparisons between algorithms (since all implementations were developed by the same team). Of course, benchmark cases also use the standard implementation, but, honestly, it was not expected to exceed their performance. These implementation are:

- The cuckoo hashing algorithm [12], being implemented in C++, Java, and Python to compare the same algorithm in different languages. This algorithm was chosen mainly because it is supposed to use two hash functions. In fact, it is not practical to do that and keep using the usual design; for instance in Java, it is possible to program a custom `hashCode` method but there is no other standard method to provide the second implementation. This algorithm is implemented using **two ranging-hash functions** and thus, the resulting implementation is fully compatible with the standard design.

- Two dummy structures: an array adapter and a sequence-of-pairs adapter have been implemented in C++ modeling the map concept. The first is an array, it uses the keys directly as the array indexes, of course, it can only be used in cases A and B since in those cases the keys are consecutive integers and, knowing that, it is possible to skip the hash functions. The array adapter is expected to be the best solution (when applicable). The second is a sequence; in fact, a `std::vector` of pairs $\langle k, v \rangle$, where pairs are searched one by one along the sequence when looking for a key. Obviously, this is expected to be the worst solution and, in fact, is only used for small sized cases.
- A custom implementation of an Hashed Array-Mapped Trie (HAMT) [13,14]. This data structure combines the properties of both tree-like structures and hash tables. For such goal, likewise other hash table-like structures, it takes a $\langle k, v \rangle$ pair and hashes its key in order to compute the location of the value within the underlying structure, which is a custom retrievable tree, also known as trie [6], prefix trees or digital trees. Due to its trie nature, the resulting hash is splitted into several prefixes which will be used to traverse the tree and locate the node corresponding to the key. In our implementation, this size is set to 6 in order to force a size of $2^6 = 64$ bits on several data structures used in the trie nodes whose memory size depends on the children's cardinality, such as bitmaps. This way these structures fit perfectly on a 64-bit computer word. A remarkable feature of the HAMT is that its nodes contain no `null` pointers representing empty subtrees. This allows it to optimize memory usage and only allocate pointers representing existing child nodes. Although the original design suggests several tweaks to optimize performance, those have not been included in our implementation.
- An implementation of a hash map tree in C++ to compare its performance with the solution used in Java, and
- As far as we know, an equivalent of the algorithm used in Python, written in C++ to compare the algorithm in fairer conditions.

All cases have been executed in 3 different computers with similar results. That is, the results are qualitatively equal, but with some offsets due to computer specifications. The actual results published in this paper and the software repository are taken using a Intel(R) Core(TM) i5-8400 CPU @ 2.80GHz with 6 cores, with the following cache memory, L1d: 192 KiB; L1i: 192 KiB; L2: 1,5 MiB; L3: 9 MiB; (from GNU/Linux command `lscp`), and 31GiB of RAM memory (from GNU/Linux command `free -h`) running a GNU/Linux Ubuntu. Since the total cache memory is important for later discussion, the total cache is: 11,136 KiB (that is, 11,403 Kb), note that the GNU/Linux command `time` specifies the output memory to be Kb (It might be that Kb is used as 1024 bytes, unfortunately, there is no way to be sure).

Software versions are: Ubuntu 20.04, gcc 9.3.0, OpenJDK 17.0.1 (64 bits), and python 3.8.10.

## 3. Results

To analyze the results, a lot of graphics have been generated automatically and reviewed. Before doing so, some comments about the programming of cases will be drawn, especially about interfaces.

### 3.1. The Maps and Their Software Interfaces

After writing about 48 programs using the map interfaces in Java, C++ and Python, there are a few comments that are worth a mention from the experience (just personal opinions here). First one, it is very easy to write programs in both Java and Python and well, not so easy in C++, no surprise here, but the interesting point is that it is really easy to translate from one to any of the others, so after some tries, the process gets established as: write in Java, translate to C++ and then to Python.

Obviously, the main concepts about hash tables and their use are the same in all of them, but it is a bit annoying that none of them name any method like the others, and,

well, this is quite a surprise because, in fact, they are the *same* methods. However, even if methods are the same they are a few important differences in programming in them:

- There is a curious difference in the hash function results. While C++ returns a `size_t` value (typically 64 bits sized), Java returns an `int` (32 bits) and in Python it seems to vary around 32 to 64 bits (using `sys.getsizeof`). As already mentioned, C++ and Java return the same integer when doing the hash of an integer but Python computes something else.
- Of course, Java and Python have memory management and C++ does not. Therefore, programming is slightly easier, but not always! If needed it is possible to use smart pointers in C++ to provide memory managing and similar behaviors, that is, smart pointers are somehow equivalent to Java or Python references to objects. In fact, the implementations of the *cuckoo hashing* and *HAMT* uses smart pointers. On the contrary, there are somethings you could do in C++ and you simply could not do it in any of the others.
- Generic programming is quite different from one to the other. The easiest and the most expressive but with the less boilerplate code is Python (by far) since, in fact, any method or function behaves as a template and in addition to the "magic methods" provide a straightforward solution to overload operators. Maybe C++ is more powerful due to the general features of the language, but it is quite verbose and sometimes very difficult to check or debug. Finally, generic programming in Java is less flexible than in the others.
- Finally, there is another curious difference in the interfaces. While Python and Java return the old value when removing an element from the hash table, C++ does not. So when the removed item is going to be used after removal, two operations are needed: retrieve the element and then remove it.

### 3.2. Results from the Benchmark Cases

The description of the cases highlights the most important results. Only a few graphics have been selected to be included in the paper, but they are all available at the software repository.

### 3.2.1. Cases A and B

These cases are designed to find the influence of ranging hashing and qualified collision in the different options. The influence is evident, for instance, Figure 1, shows the lookup speeds against the $\lambda$, that is, the value that multiplies all consecutive integers before using them as keys. The values correspond to the standard implementation of a hash table in C++, namely `std::unordered_map`. The speed is ranged from nearly $4 \cdot 10^4$ loop iterations per millisecond to under $3.5 \cdot 10^4$, so there is enough difference to infer that there is an influence but not a clear tendency, except the fact that the lowest value is at $\lambda = 13$, a prime number.

In Java, the same behaviour is observed, but lower values seem to take place at even numbers, specially at those that are powers of 2. For instance, Figure 2 shows the building speed vs. $\lambda$ using the OpenJDK implementation of the class `HashMap`. Although the speed values are different in the three speeds of the case A: building, lookup and removing, the curves are very similar for the three steps.

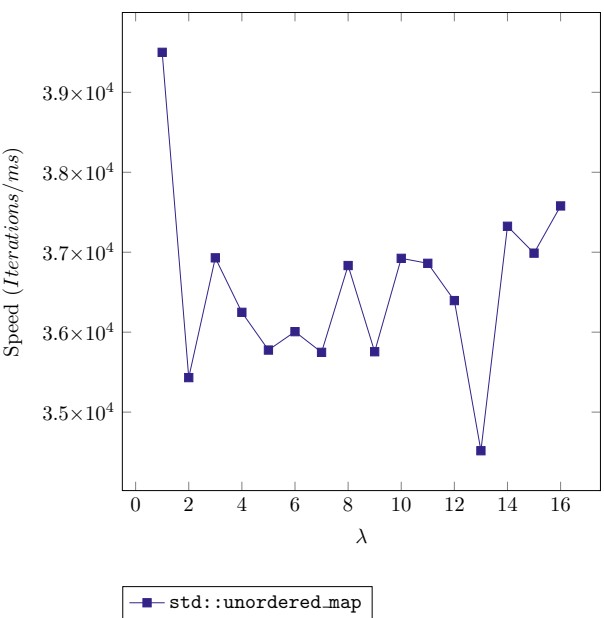

**Figure 1.** Case A. C++ `std::unordered_map` lookup speeds ($size = 2^{20}$).

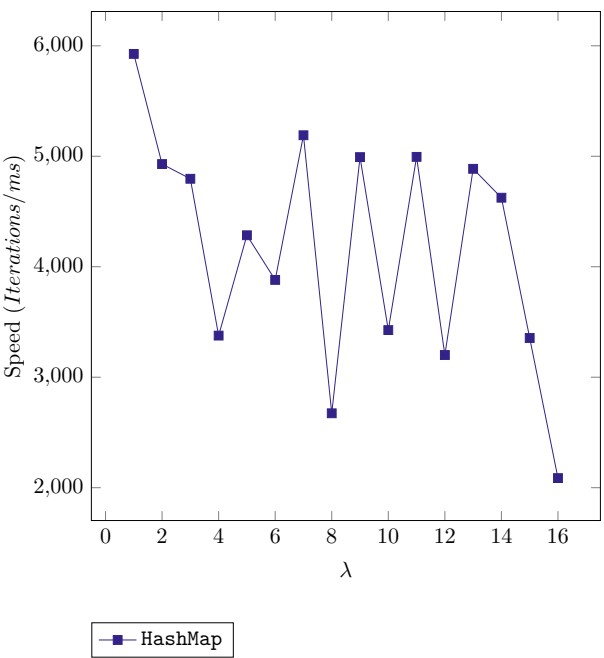

**Figure 2.** Case A. Java `HashMap` building speeds ($size = 2^{20}$).

The lowest speeds are for Python. Its curves are similar to Java, with lower performance at even numbers. This case is also computed using higher values of $\lambda$, following powers of 2 beginning at 16. The Figure 3 shows one of these samples: the removal speed vs. $\lambda$ for the Python standard map: the dictionary (`dict` class). In all programming language a clear minimum speed could be observed based on these graphics: 64, 128 or 256 are typical values, but they are not the same either among languages or case steps. For instance, the minimum building speed in C++ is at 64 while for removal speed is at 256.

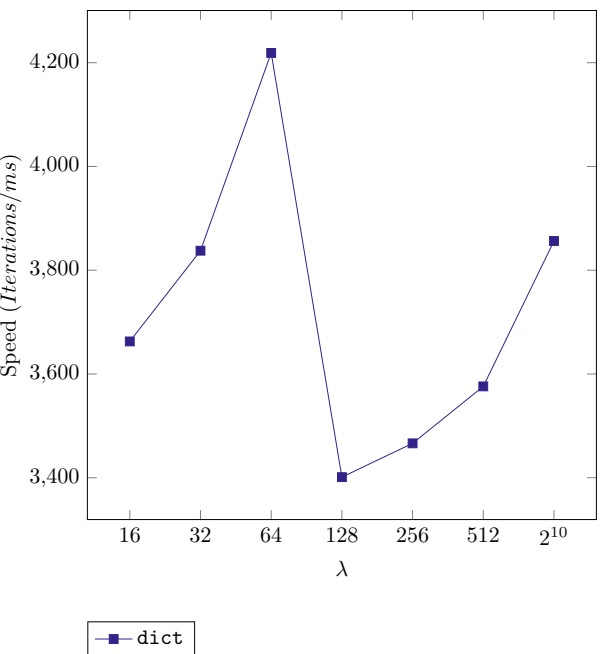

**Figure 3.** Case A. Removal Speed in python `dict` ($size = 2^{20}$).

Case B is designed to evaluate the combined influence of $\lambda$ and *size*. It is essential to state that the expected theoretical performance should be constant speed when the size changes. In this case, the focus will be on the lookup step, although all step speeds can be reviewed using the resources located in the repository. The samples in these cases have been parameterized with $\lambda = \{1, 32, 1024\}$, thus avoiding the worst values obtained from the previous case and therefore producing a fair comparison between languages. In this case, a lot of samples have been executed, since almost all data structures, standard or not, have been used in this case. All the graphics in Case B have 3 different curves for each data structure, one for each value of $\lambda$. The speed is displayed vs. the size (number of inserted elements). Both axes have logarithmic scale.

For instance, the Figure 4 displays the data for the `std::map` in C++. This data structure is a tree and the results are not a surprise: its performance decreases with the size (as expected) and $\lambda$ has very little influence since there is no hash involved.

However, displaying the same graphic with `std::unordered_map` yields surprising results. Figure 5 shows a sharp decline in the standard C++ hash table starting at $2^{17}$ and well, that is not expected because theoretically the speed should be almost constant, that is, it should keep its value about $8 \cdot 10^4$ iterations/ms as it does until that limit. On a first thought, the number of collisions could be blamed for this strange effect, so it was decided to get another reference of complexity $\mathcal{O}(1)$ and that is the reason for the array adapter. Since in this case the keys are consecutive integers, they could be used directly as indexes in an array, and due to the flexibility of C++ templates, it is possible to program such a not-so-practical data structure without any problem.

Anew, the expected result should be constant speed against the size. But Figure 6 shows once again unexpected results. Nearly at the same value, $2^{17}$, the speed is reduced dramatically, almost by one magnitude and there are no collisions in this data structure, only an array. We will discuss it later, but the fact is that in every data structure or programming language the performance degrades when some limits are reached.

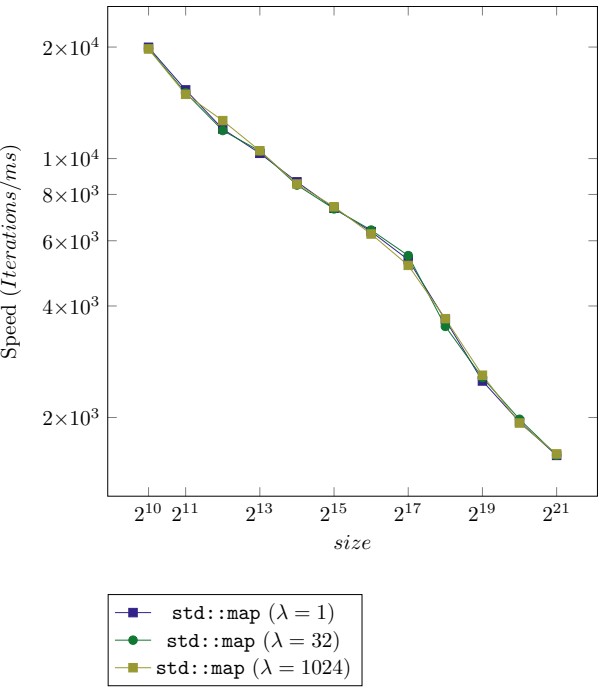

**Figure 4.** Case B. `std::map` lookup speeds.

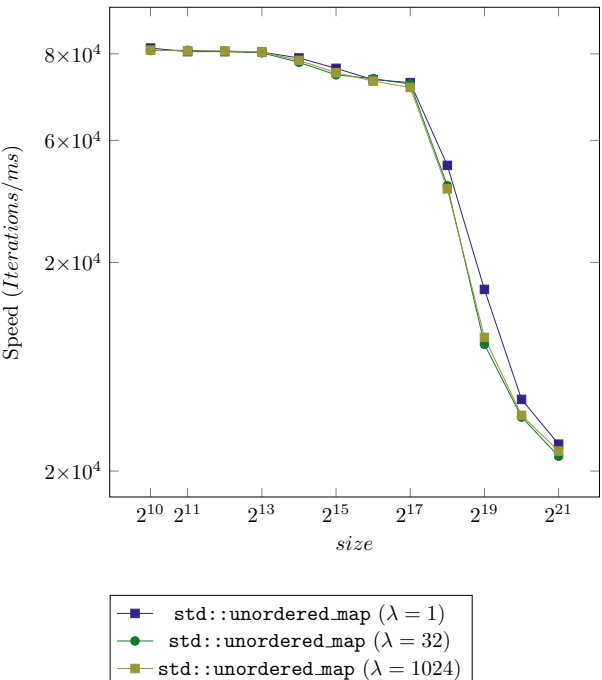

**Figure 5.** Case B.`std::unordered_map` lookup speeds.

On the other hand, the influence of the $\lambda$ parameter is far less important as the effect of the size. Thus, it is possible to keep only one of the $\lambda$ values and display some simpler graphics in order to compare the performance between programming languages and data structures. For instance, Figure 7 displays the performance of several options in C++ and Java, whereas Figure 8 shows some others with less outcome, including Python `dict`.

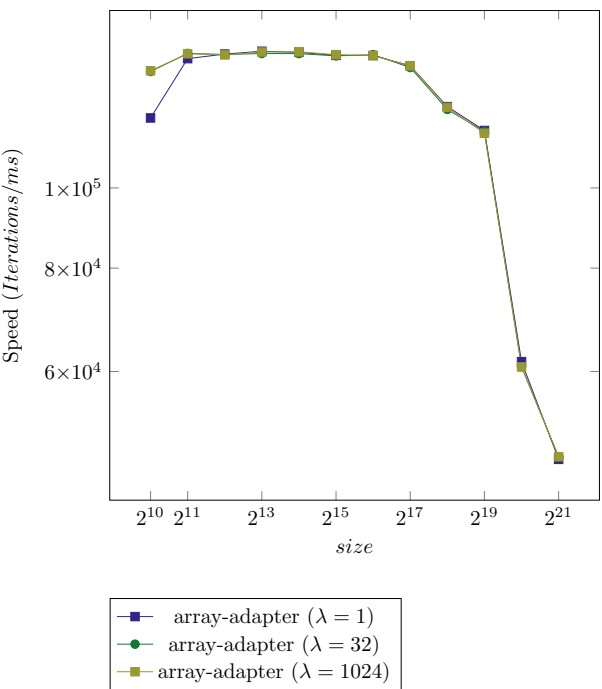

**Figure 6.** Case B. Array-adapter lookup speeds.

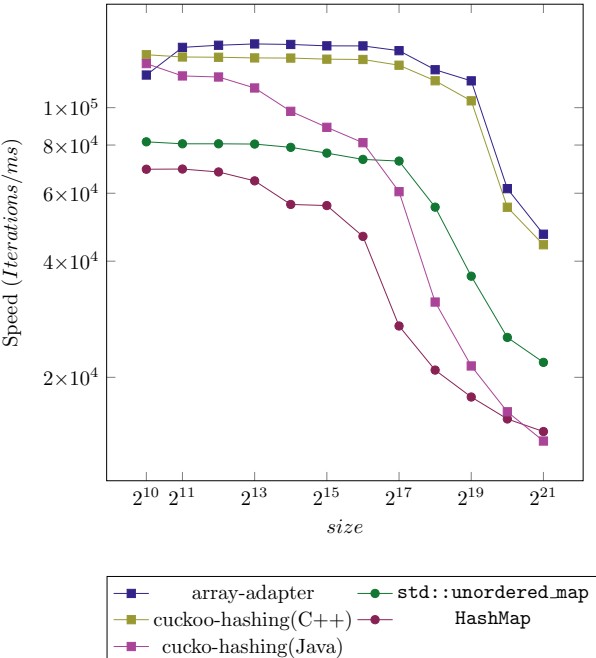

**Figure 7.** Case B. Lookup speeds ($\lambda = 0$).

The interesting fact in these graphics is that C++ is not always the fastest. It is almost the fastest in every data structure, but for instance, the implementation of cuckoo hashing in Java beats the `std::unordered_map` until size $\simeq 2^{16}$ even if it is not particularly refined. The other interesting aspect is that cuckoo hashing has an extraordinary performance. Both implementations in Java and C++ get very high speeds, although the implementation in Java degrades more abruptly than the rest. Even the worst, the Python `dict` overcomes the implementation of the `std::map` at high sizes.

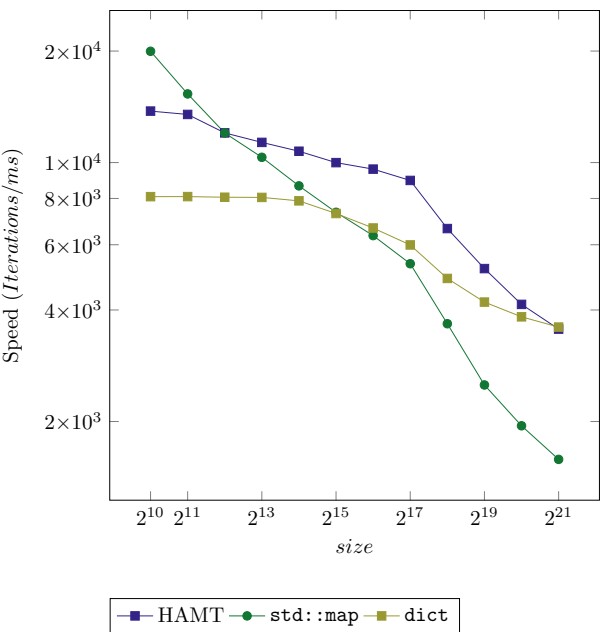

**Figure 8.** Case B. Lookup speeds ($\lambda = 0$).

Given those remarkable results, the rest of the cases try to confirm these results, generalize them, or figure out some explanation.

### 3.2.2. Case C

This case is designed to evaluate the performance of the operations of insert/removal vs. the plain access to an element. All operations take place around the fixed size, but it is allowed to remove up to 10 elements; also, insertions and removals are alternated randomly. There is a parameter that holds the chance to either run an access operation or an insertion or removal: *prob*, which is given the values: 0.1, 0.5 and 0.9 to compare speeds when the predominant operations change from one to the others. When *prob* = 0.1 the predominant operations are the insertion/removals.

The tendencies in this case are not very similar, some alternatives like C++ `std::map` does not show a lot of differences, but hash table-based solutions tend to show a slight decrease in speed when the insertion/removals are involved. For instance, the `std::unordered_map` curves are shown in Figure 9. Since the lowest speed is at *prob* = 0.1, that means insertions and removals are slightly slower than lookup operations. The results in Java are similar, but in Python (Figure 10) it is possible to observe a clear gap between the curves with *prob* = 0.5 and *prob* = 0.1, indicating that the difference between them is more important.

### 3.2.3. Cases: D, F and G

In this subsection, all cases involving *string* keys will be analyzed together. In fact, all three cases yield similar results, the curves in them have similar shapes and differ slightly in the values. Apparently, having keys of the same type is an important factor in the evaluated speeds. That said, the speeds in Case D are only a bit faster than in the rest of the cases. This is expected, since Cases F and G are more complex and each iteration contains more operations apart from the ones related to the maps. In addition, both of them consume more memory.

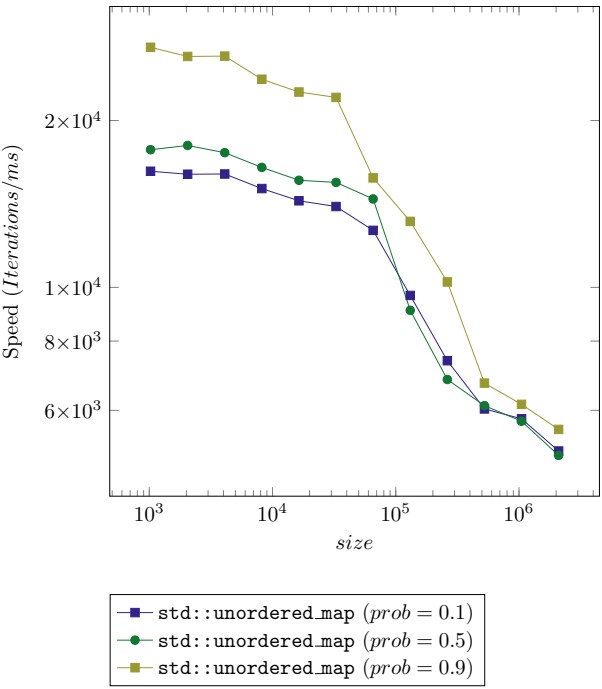

**Figure 9.** Case C. Insertion/Removal vs. Access Speed.

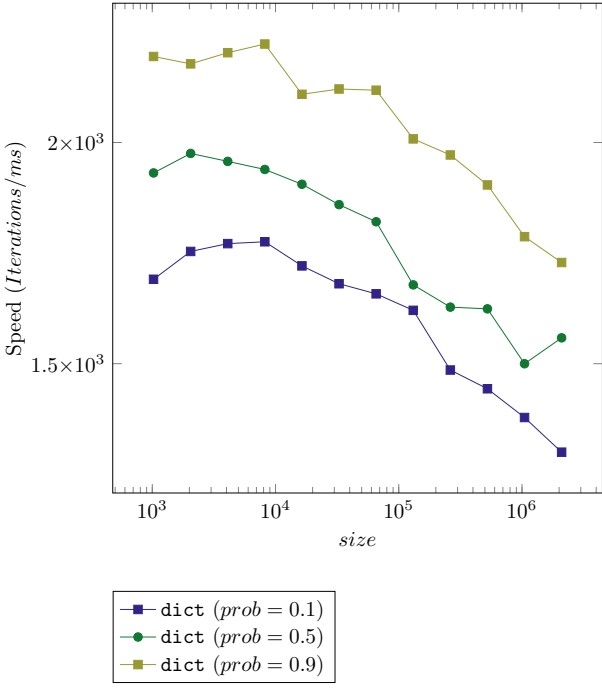

**Figure 10.** Case C. Insertion/Removal vs. Access Speed.

In these cases, as in previous ones, the building speeds are irregular, especially while comparing between programming languages. Note that there is no intention to provoke collisions: the strings are generated from numbers that all, they do not have any specific feature to produce the same hash or ranging hash. As an example, Figure 11 shows the building speed in Java standard maps: `HashMap` and `Hashtable`. In the implementation done in the context of this study of cuckoo hashing, as it seems, the speed increases with the size in the implementation of OpenJDK, probably an indication that there are some other unknown factors with influence here. As said, cases D and G have similar behaviors.

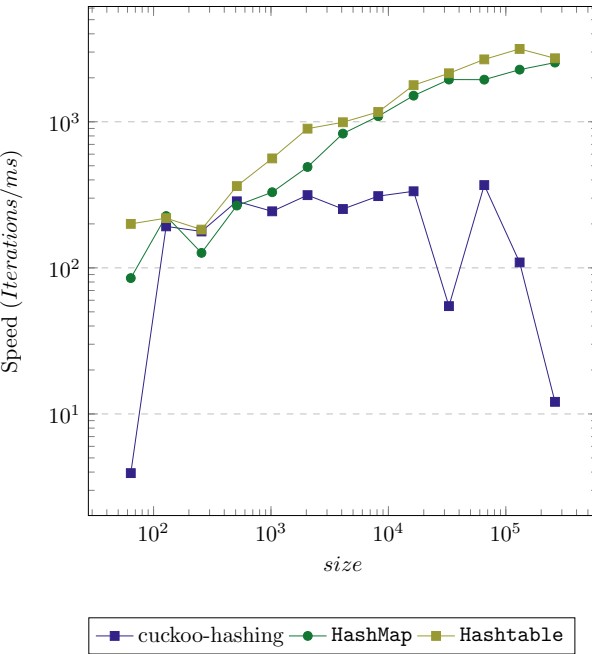

**Figure 11.** Case D. Building Speed in Java.

Regarding the lookup speeds, they also show the same decrease tendency in every data structure or language, in the standard implementation and in the others. Figure 12 shows such a tendency, being not so steep as in previous cases but still clear. In fact, this figure is quite interesting because the curve slopes are similar in all of them. That is alright, but the red line is the `std::map` whose lookup method is supposed to be $\mathcal{O}(\log N)$ while the others are hash tables with $\mathcal{O}(1)$ complexity. Note that both axes have logarithmic scales to emphasize the details. Without using the logarithmic scale in the horizontal axis, the points at low values will be too close and without it in the vertical axis the differences between hash table speed values will be minimal.

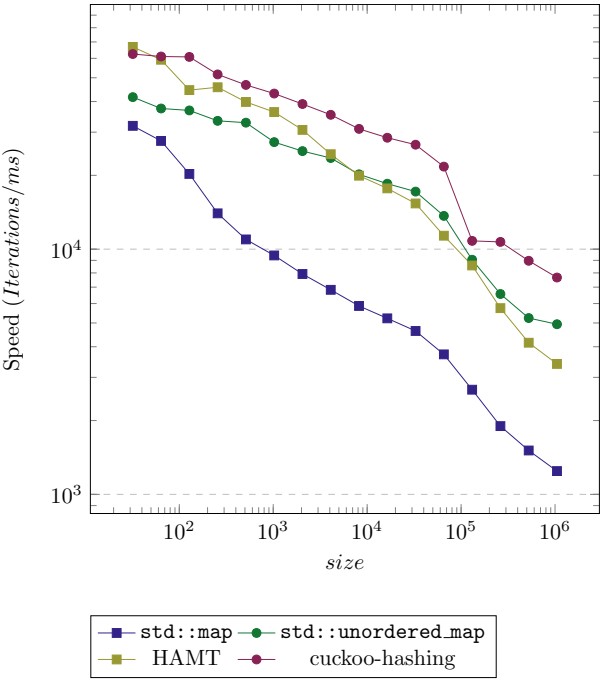

**Figure 12.** Case F. Lookup Speed in C++.

In terms of performance, another interesting aspect is memory use. Although memory availability is not a problem in modern computers, it looks like an interesting thing to measure. Maximum memory used by the programhas been measured using GNU/Linux command `time` (it is not the embedded *Bash* command time as they are different; in fact, the one shipped with Bash does not measure memory use). Figure 13 shows some characteristic examples. The most important fact here is the large amount of memory consumed by Java at low sizes, although it seems to have no relation to the memory used directly by its data structure, since its value keeps a constant level until high size values. C++ alternatives use less memory by far, being Python in the middle, except at the highest size values. It is important to note that memory use in both Python and C++ are below the available cache memory (in the computer where data was taken) for a while.

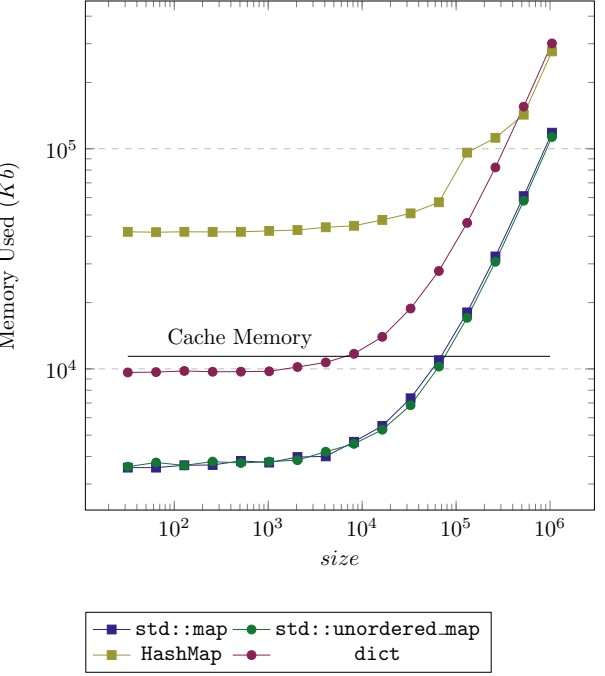

**Figure 13.** Case F. Memory Use.

Cases D and G are very similar, both use *string* keys, both have random float numbers as values, and both of them compute the sum of the values. The difference is the size in both keys and values. In case G, keys are twice the length of keys in case D and case G values are 8 numbers instead of just one. Figure 14 displays both speed and memory use for C++ `std::unordered_map` in cases D and G. Black line is the total of the cache memory according to the computer specifications. There are two interesting points in this graphic:

1. At low size, there is hardly any difference in the speeds, but the number of operations are not the same.
2. Case G uses more memory than case D (obviously), both of them use progressively more memory, eventually, both get beyond the cache limit and in both cases the speeds decrease more abruptly at those points.

Unfortunately, this analysis could not be repeated for Java because Java uses far more memory (above the limit of cache memory) at low values of the size and so it is not possible to find the crossing point. Even as Python uses less memory, it also uses a lot of memory and it is not possible to attribute clearly the memory use to the data or to the Python interpreter.

### 3.2.4. Case E

Finally, case E introduces two characteristics in the benchmark cases: (a) it avoids random data, (b) data are *sorted* to some extend. Since the keys are points in an helix and generated by means of the parametric equations, the Z coordinates can be used to sort

keys, so that, the keys could be considered sorted according to that criterion. In fact, that is the sorting criterion used in C++ std::map. On the other hand, there are no default implementations of the hash function except for Java. Therefore a very basic hash function has been implemented in C++, it just computes the hash of the 3 float numbers by turn and sum to the previous result and multiply by 31. While in Python, the hash is produced by transforming the 3 floats into a tuple and them computing the default hash for tuples.

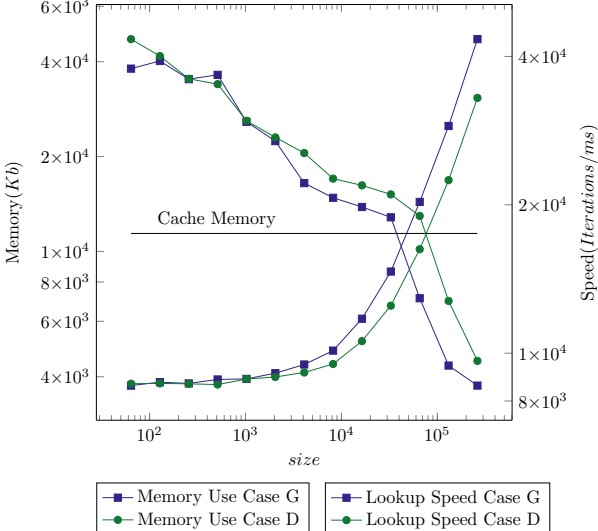

**Figure 14.** Cases D and G. Memory Use.

There are no surprises in the results here. Of course, the std::map is favored by sorted keys as it can be observed in Figures 15 and 16. The first shows that the std::map, that is a tree achieves the fastest building speed, the second shows a narrower gap with respect to the rest alternatives, it is still the slowest but the difference is not so high. On the other hand the hash tables maintain their observed tendencies, all of them have decreasing speeds when the size grows.

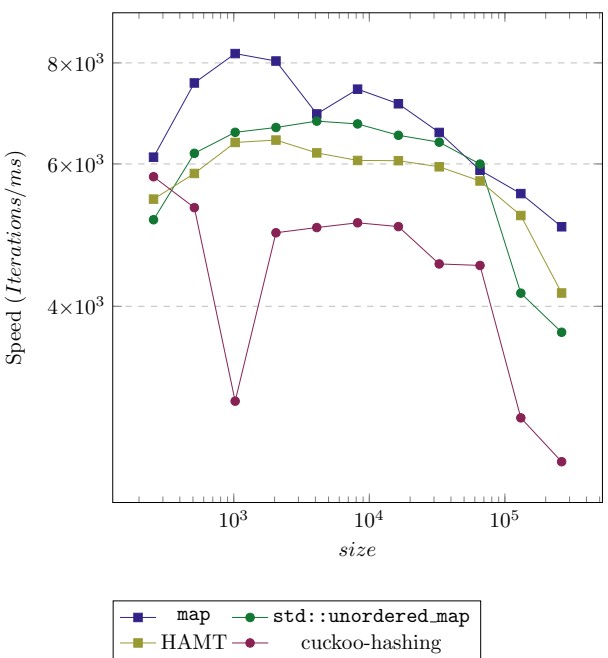

**Figure 15.** Case E. Building Speed in C++.

Another significant fact can be observed in Figure 17. In this case, the fastest implementation is Java `HashMap`, clearly above C++ implementation. The origin of this good performance might rely on the hash function, being the C++ hash a custom function, it is probably worse than the standard implementation in Java. However, this is interesting anyway since in the rest of benchmarks, C++ was always the fastest.

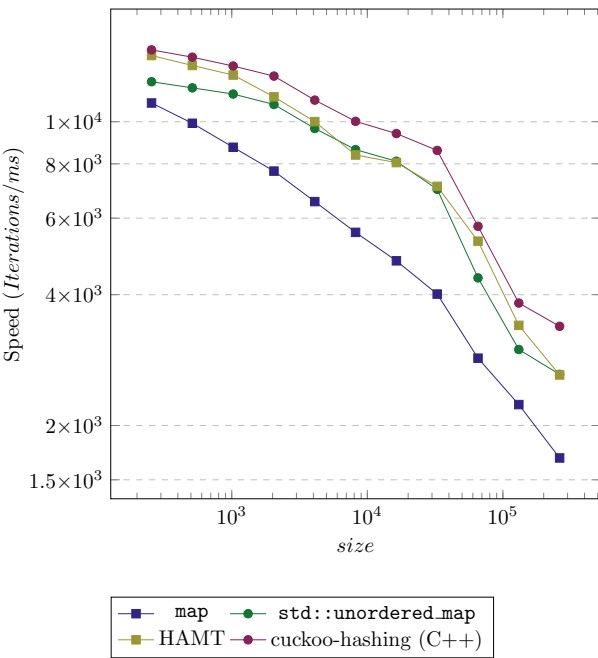

**Figure 16.** Case E. Lookup Speed in C++.

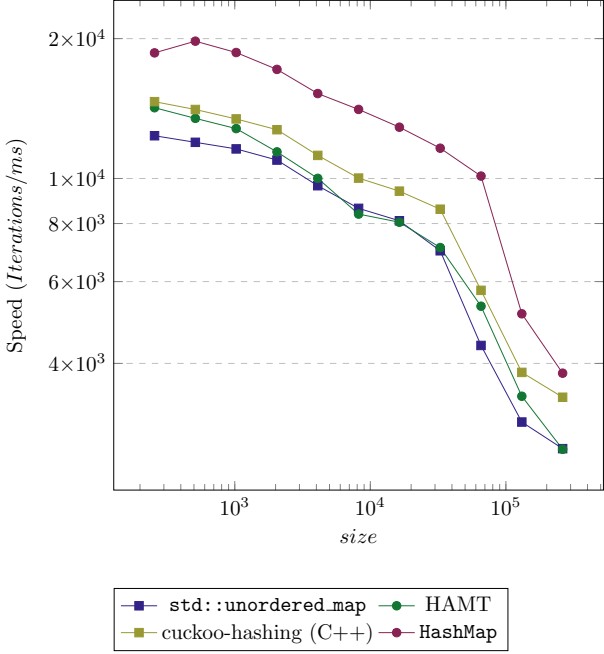

**Figure 17.** Case E. Lookup Speeds.

## 4. Discussion

The objective of this study was to assess how hash and ranging hash functions could cause a weakness in the standard implementation of hash table. In fact, the results of the case A show that weakness, although it has less importance than expected.

Meanwhile, some unexpected results were found by varying the size, the number of elements, that is, the N, by its common symbol in complexity analysis.As already said, this finding reoriented the survey in order to isolate the influence of the size. One of the main decisions was to design the benchmark cases into two separated steps: building the map, that is the insertion of all elements, and running something else that involves lookup operations.

The critical feature about the lookup step is that its loop has a fixed number of iterations. In addition, it makes some kind of accumulative computation to avoid being optimized out by compilers. That way it is possible to measure the times with high reliability, in fact, since the elapsed times are taken before and after the loop and the number of iterations is above hundreds thousands of times; the measure is quite deterministic and the error is very low. Nevertheless, strictly speaking, the measured times include more operations apart from looking up the hash table, but all of them are simple operations and none of them have anything involving a *size*. Note that the number of iterations is the same for all sizes.

After these adjustments, it was clear from the facts that look-ups were not constant time. In all languages, in all implementation, sooner or later the look-up speed drops steeply at a size value. That is quite unexpected, all programmer manuals state one way or the other that (average, amortized, . . . etc.) constant time is guaranteed. None of them mention any limit for that property, but benchmark provides another evidence, that a limit exists.

What is the cause of the behavior? Well, complexity analysis is about operations, it counts operations, the operations could be observed and count in source code, but what about memory use? Theoretically, memory use is evaluated in terms of memory limits, that is, nobody care about memory use with respect to performance unless it goes above the hardware limit (the RAM memory). Well, facts establish that memory use matters for algorithm performance. Our last benchmark tries to identify the performance dropping point, and there it was, that point is around the size where the data structure fills the L3 cache memory. That is something expected as cache memory is meant to do that.

So, could hash tables be blamed for dropping their performance? The answer is no. As already said and shown in Figures 6 and 7, the performance of an array also drops at a given size, so the *fact* is that no present or future hash table could achieve an *empirical* time complexity of $\mathcal{O}(1)$ because in their actual design they use arrays (the *buckets*) and reading a value from an array is not constant time, that is:

$$array[i] \textbf{ is not empirically } \mathcal{O}(1) \tag{1}$$

This is a consequence of the real architecture of modern computers. In the theoretical analysis on complexity, memory is modeled uniformly, but somehow that model is obsolete. Therefore, theory fails to estimate algorithm computing time cost because there is a substantial difference between the theoretical model and the real hardware.

In addition to that main finding, the extraordinary importance of hash and ranging hashing was found again and again. For instance, it is really interesting that Java implementation goes faster than C++ in case E, Java has a reputation of being slow, but it wins clearly, and that is an unarguable evidence of the importance of hash function and it also tell us that the performance has no absolute reference between programming language, implementation matters.

It is also worth mentioning that Java uses more than one CPU kernel. That is an interesting feature since there is only one thread in the program sources and somehow the execution in Java gets parallel processing. Apart from the system monitor, the parallel execution could be observed by means of the measured times using the GNU/Linux `time` command: the sum of "User Time" and "System Time" is higher than the "Elapsed Time". That feature could explain some speed increases at medium size values.

Finally, a note about the magnitude order of speed. Roughly speaking, lookup speeds are about $10^4$ iterations per ms. That is about $10^7$ iterations per second, even Python, the slower alternative, get above $10^6$ iteration per second in most cases. Considering that not

so long ago, computer performance was measured in MFLOPS, these speeds are really high. In addition, in most benchmark cases, the loops not only call the lookup operation on the map, but they also do some other operations (maybe simple operations, but extra operations anyway). Considering that each look-up operation have to call the hash function, resolve eventual collisions and so on, the speed is very high. As a consequence, using a performance criterion in order to choose one option over the rest should be only done when performance is the most critical issue and lots of operation of these kind will be expected.

## 5. Conclusions

Complexity analysis can not fully estimate computing times in modern computers. The evidences found in the benchmark cases show clearly that prediction fails at high values of the data structure sizes.

Since complexity analysis does not take into account the operations for memory management, it overlooks the impact of data transmission between RAM memory and cache memory. While a simple operation on any data is extraordinarily fast (if the data are already in cache memory), the time for reading RAM memory is much longer. Therefore, complexity analysis ignores operations that have a high impact on performance and counts other operations that are negligible with respect to computing time. It is like estimating the mass of a group of animals counting ants and missing elephants.

Indeed, an analysis that is unable to estimate its target result is hardly useful and that is bad news. Complexity analysis has been a decisive tool to improve algorithm efficiency and, to some extent, will be important in the future. However, at this moment, its computer model is no longer valid. It is supposed that the computer memory is uniform and this supposition is far from being accurate. Cache memory has been a key factor to improve computer performance; thus, while the theoretical model only has one uniform memory, real computers have two types of memory that are extremely different in features: an almost unlimited, slow RAM memory and a scarce, super fast cache memory. Unfortunately, the usual high-level programming languages have no capability to operate or manage these types of memory.

The impact on software performance of cache memory is considerable, concepts like: cache alignment, memory locality, cache miss, or the importance of the cache size in CPU performance are typical topics these days. The introduction of these concepts is a clear clue of the necessity to review the theory to estimate algorithm performance. Since memory management has an impact in computing time, new algorithm designs should carefully consider: memory layout, memory use, and memory alignment along with the number of operations. The importance of these issues will be increase with data size, being a key factor for scalability.

On the other hand, with this results, someone may argue that the predictions are accurate at low sizes, but they are not. Along benchmarks, the C++ `std::map`, being a tree, has been used as a performance reference, the complexity of its operations are reported as being $\mathcal{O}(\log n)$ instead of $\mathcal{O}(1)$, so that is the reason of being slower. Well, definitely that does not have consistency. At low sizes, we can say at $size = 64$, the value of $\log n$ is so low that it can hardly explain any difference in computing time. Therefore, again, the cause of the speed difference should be looked for in another place. Most probably, the different use of L1 and L2 cache memory, due to data locality, could be blamed for that, but finding sucha cause could be the objective of another work.

It is clear that memory use is a critical issue, not to be compared to the almost unlimited resources of the RAM memory, but with the scarce limit of (L3) cache memory. That is an important fact since some programmers think that it is better to use memory instead of doing operations and that, definitely, will hold for small sizes, but will not scale well at larger sizes. For example, we can say it is possible to save one byte per data item (without compromising alignment and other low level memory restrictions, including those related to the use of L1 cache memory); if the number of data items will exceed one million of

elements, do it, it will save about 1 MiB of memory at L3 cache memory and it will be worth it.

Finally, it is our opinion that the documentation and manuals about hash tables should change. In their actual wording, they do not mention any limits or restrictions to "constant time", but we think we have proven the opposite with enough evidence. Arguably, someone might say that the documentation is correct because complexity analysis is independent of real hardware, but, at least, this is misleading. No one reading the documentation of a data structure is going to run his program in a theoretical computer, and, therefore, any one using a hash table will expect the same performance at any data size.

**Author Contributions:** Conceptualization and methodology, S.T.-F.; software, all three authors; validation, all three authors; writing—original draft preparation, S.T.-F.; writing—review and editing, all three authors. All authors have read and agreed to the published version of the manuscript.

**Funding:** This research was partially funded by FUNDACIÓN PARA LA INNOVACIÓN INDUS-TRIAL (FFII).

**Data Availability Statement:** The full results of execution times and memory use, with the reported computer, could be found at https://bitbucket.org/b-hashtree/b-hashtree/downloads/, (accessed on 10 February 2022), filename is *CasesTSV.tar*. It also contain some PDF with more figures of benchmark cases.

**Conflicts of Interest:** The authors declare no conflict of interest. The funders had no role in the design of the study; in the collection, analyses, or interpretation of data; in the writing of the manuscript, or in the decision to publish the results.

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
