# Peer review of "Key Concepts, Weakness and Benchmark on Hash Table Data Structures"

_algorithms, doi:10.3390/a15030100_

Round 1
Reviewer 1 Report
Very interesting article describing empirical testing of data structures commonly used in application development. The article proposes tests to examine various aspects of these data structures. The evaluation of the tests is described both in the article itself and in the supplementary materials. Unfortunately, the way these tests are presented is also the biggest weakness of this paper.
Comments
line 81 - The authors neglected to point out here that tree-like structurings require that an ordering session be defined on the key set K. This is generally not necessary for the remaining Standard sequences and Hash tables.
Section 3.2 in general
The main problem of the paper is mainly the processing of graphs and labeling of variables and names of data structures that occur in the text.
I think it would be a good idea to introduce some single-letter designation for shift and size, say \alpha and \beta. It's entirely up to the authors of the paper. And thus clearly separate the word shift from the shift shift. For example, on line 331, the symbol shift is (mathematical italics) but size is already a standard word. I think it would be better to write "influence of the shift $\alpha$ and size $\beta$." Figure 1 caption - "Size=2^{20}". The word size should be in a mathematical font and lowercase. Or \beta should be used.
Another problem is the typesetting of data structure names. On line 314, std::unordered_map is typed, which matches the caption of Figure 1. But in the legend of the graph it is just unordered-map. On line 291 it says "HAMT uses smart pointers". Uppercase HAMT. Caption of figure 8 - just "hamt", Figure 17 "Hamt".
Numbers in text and graphs. There are at least three different styles of writitng numbers:
a) Just integers 10, 256 or decimal numbers like 0.9 these are ok.
b) Scientific, exponencial notation like 2^{17}, these are also ok.
c) Semilogarithmic numbers with e letter, see line 314, 315. Why 4e4 notation is used instead of $4 \cdot 10^{4}$? On line 387 there is $prob=0.5$, but Figure 9 and 10 uses $prob=5e-1$. This kind of notation is very inappropriate and also confusing, expression 5e-1 should be interpreted as "five times Euler number minus 1"? No it is just 0.5!!!!
And now the axes of the graphs. Figure 1 - very nice graph! Y axis has values from 35,000 to 40,000 in scientific notation. Figure 2 and 3 - nice graphs. Y axes can be converted also to scientific notation, but it is not necessary. Figure 4 - y values are in semilogarithmic notation, why. There is no reason to switch tne notation. Values are very similar to Figure 1, and scientific notation is used there. And x axis of the graph uses this notation. The same holds for figures 5, 6, 7 and 8. Figure 9 introduces even fractional exponents! When I am reading this graph, I cannot clearly imagine what value is $10^{3.8}$. Yes it is just under ten thousand. May be 8 thousand? No, calculator says that 6309,5734448019324943436013662234 is the correct result :-)
Please do not use semilogarihmic notation and fractional exponent at all!
Some graphs show the number of iterations per millisecond on the y-axis, some the number of operations per millisecond. I'm not sure what the difference is.
Also colors for data series can be improved. Please consult colors with Tol P.: Colour Schemes. SRON Technical note SRON/EPS/TN/09-002, 2012. This technical note provides very nice color for data series that can be very clearly recognized even on data projector or B&W printers.
The size of graph also should be changed. Pages like 13, even 15, 17, 23 look weird.
line 450 - Should be "Python" instead of lowercase python.
line 405 - "cases G and G have similar..." probably some typo error.
Author Response
Thanks very much for your review. We think it has been very helpful and constructive.
I think we have addressed all your comments and fixed them. Just a short note about 3 things:
1) Number of Operations was a typo. Actually, "NumberOfOperations" is the name of this parameter in the source code but then we realized that "Number of Iterations" was more precise and we use this name in the text.
2) We change "shift" to lambda, that is a similar situation, we used shift in the sources, but definitely, lambda is better. On the other hand, even if "size" is both a parameter and a common word we think it is better to keep it because the parameter and the usual meaning are exactly the same.
3) We think the final format will take place after revision, so we expect that figures will be positioned and scaled to a suitable size at that stage.
Reviewer 2 Report
The article deals with the implementation aspects of hash tables in various programming languages and under various dataset sizes. The main target of the research is to locate inconsistencies in the performance of the various hash techniques between theory and practice, and to provide valuable insight to prospective users of this technique in various software applications. In order to achieve that the authors have developed benchmark cases designed to be randomized, self-tested, and representative of a real user cases.
The outcome of their various experiments was quite interesting depicting that there are major inconsistencies between theory and practice due to various optimizations (like cache maintenance) in real software systems that seems to grow larger when the dataset is large. On the other hand, for small datasets practical performance seems to be the same even for data structures with different asymptotic complexities.
As a whole, the article is interesting practical research on an aspect that is quite common in software applications, that of efficient hash table implementations. On the other hand, the novelty of the present work seems to be limited since it does not contain new algorithm components to the community. I vote for acceptance subject to revision where in the revised version the authors should try to enhance the novelty of their work. A good direction to follow would be to distill from their experiments, general directives to prospective implementers of hash tables, plus some directions to improve classical implementations.
Author Response
Thank you for your review.
Actually, you are right, we do not pretend to have any new algorithms, instead we focus on reviewing and evaluating the existing ones. On the other hand, we do not pretend neither to provide an orientation since that will be an opinion or intuition and we try to present only facts not opinions.
That being said, we have written a new full subsection and added some paragraphs at the discussion and conclusion sections. Since you and other reviewers asked for more comments (intuitions, opinions,...), we include in those paragraphs some intuitions based on what we know about modern PCs and our experience in designing and programming algorithms. And, with respect to directive, to summarize: mind the memory; memory management is, probably, the most important factor in computer performance nowadays. Unfortunately, up to date, we could not say which is the best way to manage memory... But we are working on it.
Just a final note, we have added something we overlooked to emphasize in our first version: our experiments show that reading a value in an array (that is, array[i]) is not constant time. Therefore, there is an implication here: hash tables could not be improved directly because it is not their fault, the decrease in performance is from their internal arrays.
Reviewer 3 Report
The research investigates hash table data structures; as we are familiar with, they do have to solve hash collisions. Specifically, some key concepts about hash tables along with some definitions about those key concepts are introduced, especially the characteristics of the main strategies to implement hash tables and how they deal with hash collisions. Then, some benchmark cases are designed and presented to assess the performance of hash tables. These cases have been designed to be randomized and representative of real user cases with the aim of analyzing the impact of different factors over the performance across different hash tables and programming languages.
In order to better understand the outcome of this work, authors programmed all cases using C++, Java and Python and analyzed in terms of interfaces and efficiency (time and memory).
This is a very challenging topic and area and authors tried to prove their findings. However, there are several issues that need to be answered before the paper can be published:
The contribution subsection (as well as other related paragraphs) must be rephrased in order to include the problem, the goal as well as general remarks regarding the results. Table 1 in page 6 is not enough for differentiating the proposed work from others similar.
In the introduction, authors fail to precisely include the basic intuition behind this work. They should prove the contribution in a more solid way.
Results section is just some experiments in terms of tables with proper explanation. What authors try to prove with these experiments? Isn’t somehow this contribution trivial? How are they differentiated from other works?
The authors cannot efficiently state the differences among their work and the references.
The discussion section is based on these implementations. Maybe, authors try to validate more in detail these structures in terms of their weaknesses in order for their paper to be more comprehensive.
Author Response
Thank you for your review.
Actually, you are right, there is no basic intuition behind our work. We have written a full new subsection in order to explain how our work began and why it evolved in the direction it took, but in summary, we did not pretend to write an article that makes a contribution but a survey or a perspective.
We have also rewritten and added more explanations in discussion and conclusion sections, we think that these explanations improve the general understanding of our work, its implications and our intuitions about the reason behind the experimental results.
That being said, we do think we make a contribution, only that, our contribution is bad news. We did not do anything new, but we reviewed and reevaluated statements that are supposed to be true and many people think so (including us in the past). But, by evaluating hash table data structures we find that complexity analysis is obsolete, we find that reading a value from an array is not constant time and, therefore, there is not any hash table that will be able to hold constant time at large sizes. Honestly, I do not know about any other paper that hold these statements (and provide a proof), and, even if there is such a paper, we think our paper should be published for the reason we explained in a new paragraph at the end: documentation and manual should change, they are misleading programmers about algorithm performance. Should the documentation and manual keep being the same, our article and others about this subject should be published until the "established" knowledge about this matter changes.
Reviewer 4 Report
The authors do the study with 3 languages (C++, Java, Python). The document should discuss why they have chosen these 3 languages and not others.
Author Response
Thank you very much for your review.
We have written the explanation in the manuscript (at the beginning of section "2. Materials and Methods'', as it is also said now in the footnote, we were working on other languages, actually C# and PHP for the same reason (we skip javascript because its usual environment is a Web browser and that introduces another factor to be studied or require install node.js and that is not so straight forward like the other options).
Anyway, configuring and, especially, analyzing the results takes a huge amount of working time and, in some preliminary cases, we began to find exactly the same kind of results so it would not change the main findings.
Round 2
Reviewer 1 Report
The article has been significantly improved. I recommend acceptance.
Otherwise, I think the most important message of the article is on line 574.
Nice work!
Reviewer 3 Report
Authors have addressed my comments.